# Process Intensification Strategies for Power-to-X Technologies

**Thomas Cholewa [1,2], Malte Semmel [1], Franz Mantei [1], Robert Güttel [2] and Ouda Salem [1,*]**

1. Fraunhofer Institute for Solar Energy Systems ISE, Heidenhofstraße 2, 79110 Freiburg im Breisgau, Germany; thomas.cholewa@ise.fraunhofer.de (T.C.); malte.semmel@ise.fraunhofer.de (M.S.); franz.kaspar.mantei@ise.fraunhofer.de (F.M.)
2. Institute for Chemical Engineering, Ulm University, Albert-Einstein-Allee 11, 89081 Ulm, Germany; robert.guettel@uni-ulm.de
* Correspondence: ouda.salem@ise.fraunhofer.de; Tel.: +49-761-4588-5335

**Abstract:** Sector coupling remains a crucial measure to achieve climate change mitigation targets. Hydrogen and Power-to-X (PtX) products are recognized as major levers to allow the boosting of renewable energy capacities and the consequent use of green electrons in different sectors. In this work, the challenges presented by the PtX processes are addressed and different process intensification (PI) strategies and their potential to overcome these challenges are reviewed for ammonia ($NH_3$), dimethyl ether (DME) and oxymethylene dimethyl ethers (OME) as three exemplary, major PtX products. PI approaches in this context offer on the one hand the maximum utilization of valuable renewable feedstock and on the other hand simpler production processes. For the three discussed processes a compelling strategy for efficient and ultimately maintenance-free chemical synthesis is presented by integrating unit operations to overcome thermodynamic limitations, and in best cases eliminate the recycle loops. The proposed intensification processes offer a significant reduction of energy consumption and provide an interesting perspective for the future development of PtX technologies.

**Keywords:** process intensification; Power-to-X; process integration; dimethyl ether; oxymethylene ether; ammonia; reactive distillation; adsorptive separation

## 1. Introduction

Code red for humanity was addressed in the recent intergovernmental panel for climate change IPCC report published in summer 2021 [1]. The emerging issue of climate change enforces the drastic reduction of greenhouse gas emissions. Electrification by boosting of renewable energy capacities to supply all energy demanding economic sectors is necessary to reach the defined climate change mitigation targets [2]. The production of green hydrogen ($H_2$) using renewable energy and the subsequent synthesis of chemicals in Power-to-X (PtX) processes was identified as an important pillar for sector coupling and for the transformation towards a sustainable energy system [3]. PtX is defined as the integration of renewable energy beyond direct electrification into the energy, mobility, industry and private sectors via $H_2$ based renewable energy carriers [4]. The production and import of renewable energy from locations with a high potential will be a crucial element of the future energy system and will inevitably require a suitable energy carrier. All long-distance and heavy duty transportation will rely to a large extent on dense liquid fuels in the future [5]. Regarding the German energy market, the annual demand for energy in the form of PtX molecules is estimated to range from 75 up to 500 TWh by 2050 [5,6]. Possible products of PtX processes vary from synthetic fuels such as E-diesel or E-kerosene, to common bulk chemicals like methanol (MeOH) and ammonia ($NH_3$), and further to highly processed chemicals and oxygenated fuels like dimethyl ethers (DME) or oxymethylene dimethyl ethers (OME).

### 1.1. Boundary Conditions for PtX Processes

One of the main challenges of renewable energy-based processes is the geographically dependent potential and temporal fluctuation of renewable resources and consequently fluctuating electricity production. Green $H_2$ production based on these fluctuating green electrons is technologically possible via water electrolysis, which is subsequently used as the feedstock for synthetic fuels, chemicals and energy carrier production. Since steady-state operation is the common strategy in conventional large-scale chemical or refinery processes, new operational strategies are needed. Available strategies are either the storage of $H_2$ under elevated pressure or cryogenic conditions or alternatively, to operate the downstream process dynamically, a major emerging challenge for these industries. However, $H_2$ storage remains energy intensive and expensive. For instance, the cost for storage can reach up to a 25% share of the net production cost (NPC) of $H_2$ production in small and medium size systems [7]. Furthermore, proposed large scale $H_2$ storage technologies such as salt caverns, are scarce and not necessarily available at locations rich in renewable resources [8]. In the light of the previous discussion, the latter strategy of dynamic PtX process operation can minimize and ultimately avoid $H_2$ storage allowing for significant NPC reduction. A dynamic process in the context of this work refers to the possibility to control the process conditions and plant utilization based on the availability of renewable energy. However, the dynamic operation of reactors, compressors and other unit operations exhibit challenges that require investigation and further development. For the example of heterogeneously catalyzed exothermic reactions performed in fixed-bed reactors some of these major challenges are:

(a)　the thermal instability and transient hot spot formation,
(b)　the possibly enhanced catalyst deactivation and degradation due to thermal cycling and load changes,
(c)　and the transient changes in product quality or composition and possible undesired side product formation.

Locations with high potential for renewable energy production are often located in remote areas without connection to the infrastructure available at industrialized areas or even offshore. The realization of chemical processes at remote locations without grid connection presents several additional challenges, such as:

(a)　no compensation of fluctuating electricity using grid electricity,
(b)　the elaborative production of utilities onsite using renewable resources,
(c)　the high costs of operation and maintenance,
(d)　and the limited available area for construction.

Moreover, from an economic point of view, the usage of green $H_2$ emphasizes the need for highest material and energy efficiency of PtX processes. This was supported by recent studies investigating the production of different PtX products which revealed that green $H_2$ production costs represented more than 60% of the NPC of various PtX products [9].

### 1.2. Objectives of This Work

In this work, the arising challenges for PtX processes are addressed for three important PtX products namely: $NH_3$, DME and OME. Process intensification (PI) strategies suitable for these heterogeneously catalyzed thermochemical processes are introduced. Moreover, the state-of-the-art is briefly discussed followed by the introduction of the most promising examples for improving the respective process via PI based on recent literature. The potential of PI methods for combining heterogeneously catalyzed reactions and the subsequent separation processes aiming to shift the thermodynamic equilibrium towards the desired products is identified and discussed. Emphasis is placed on quantification of the improvements obtained by PI, to evaluate the potential of respective measures.

*1.3. Background and Process Intensification Approaches*

Since the term "Process Intensification" is not defined consistently and the variety of PI approaches remains wide, a short definition and the understanding of PI in the present work is provided as follows. The motivation for the development of PI technologies is described as "doing more with less" by Jenck et al. [10], comprehensively leading to an increased process efficiency as the overall goal of PI approaches. Stankiewicz et al., define PI as "any chemical engineering development that leads to a substantially smaller, cleaner, and more energy efficient technology", which result in a "cheaper and sustainable" technology [11]. According to Ramshaw et al., PI is a strategy for making "dramatic reductions in the size of a chemical plant to reach a given production objective" [12]. Van Annalandt et al., investigated into PI for PtX products and concluded that the application of PI to the energy sector can result in a dramatic decrease in the production of waste, including greenhouse gas emissions [13].

For a better overview, the categorization of PI approaches into (a) process-intensifying equipment and (b) process-intensifying methods was proposed in the literature [14]. The process-intensifying methods describe novel process concepts, such as hybrid separations or the usage of alternative energy forms or sources [15]. As chemical reactions and the chemical reactor represent the centerpiece of most chemical processes, PI methods often focus on this process unit [16]. Most importantly for the scope of this work, the integration of several unit operations into multifunctional reactors provides a promising approach of the process intensifying methods. Multifunctional reactors combine the reaction with another unit operation that would conventionally take place in a separate process apparatus. For instance, the removal of a reaction product or by-product in the reactor can decrease kinetic or thermodynamic limitations and accordingly enhance the reaction regarding selectivity and conversion. Due to the increasing conversion, recycle streams and required equipment can be avoided, which consequently allows for a reduction in the number of process units by the integration of multiple process units within the multifunctional chemical reactor [16]. Additionally, a lower demand in utilities can offer an important benefit of a highly intensified process.

## 2. Process Intensification for Power-to-Ammonia Processes

*2.1. Background*

$NH_3$ is one of the most produced chemicals globally, with a production rate of 150 Mt/a in 2019. In addition to the current importance of $NH_3$ as a base chemical, its usage as a chemical energy storage molecule in the Power-to-Ammonia (PtA) process is discussed extensively in the scientific literature and in recent political scenarios [17]. This is due to the fact, that compared to pure $H_2$, $NH_3$ exhibits a high volumetric energy density and lower costs for storage in the liquid phase under comparatively mild conditions [18]. Thereby, either a low-temperature storage at $-33\,°C$ under ambient pressure or a pressurized storage at around 16 bar under ambient temperature is possible [19]. $NH_3$ is synthesized from $N_2$ and $H_2$ according to the stoichiometric Equation (1), whereby the synthesis conditions are defined by the reversible, exothermic nature of the reaction. With respect to the raw materials, $NH_3$ can indirectly be produced from air and water by using atmospheric $N_2$ from air and green $H_2$ produced via water electrolysis in a PtX scenario. Compared to other PtX processes, the PtA process does not require a carbon source, providing the $NH_3$ production with a unique flexibility in various locations [20].

$$N_2 + 3\,H_2 \rightleftharpoons 2\,NH_3$$
$$\Delta H^o_{298\ K} = -92.44\ kJ\ mol^{-1} \tag{1}$$

Furthermore, $NH_3$ can be used flexibly as a $CO_2$ free fuel in fuel cells [21] or combustion engines [22], as a $H_2$ carrier [23] or as a base chemical (i.e., in fertilizer production). State-of-the-art $NH_3$ production is based on fossil $H_2$ produced from natural gas via steam reforming. In this process large amounts of $CO_2$ are emitted. Consequently, the global

scale of $NH_3$ production based on a fossil feedstock constitutes almost 2% of global $CO_2$ emissions and the decarbonization of the conventional Haber–Bosch process provides an immense potential for reduction of $CO_2$ emissions [24].

### 2.2. Conventional Haber-Bosch Process

$NH_3$ synthesis is performed based on the Haber–Bosch process developed in the beginning of the 20th century. The main process units comprise the compressors for synthesis and recycle gas, the $NH_3$ reactor and the condensation unit to remove $NH_3$ from non-converted reactant gases as shown in Figure 1a. The process is characterized by its harsh temperature and pressure conditions required for synthesis at pressures between 100–300 bar and temperatures above 450 °C. Reaction conditions are defined by the commonly used multi-promoted, magnetite-based iron catalysts [25]. To reach sufficient kinetic activity high temperatures are needed. In consequence, the synthesis pressures need to be increased, to shift the chemical equilibrium towards $NH_3$ formation [26]. Due to the advantages of these catalysts, such as low costs and high structural stability under harsh reaction conditions, industrial relevance for other catalytic materials is low [27]. Increasing the production scale and a high degree of energy integration led to a drastic reduction of energy demand of modern Haber–Bosch processes. Recent world-scale plants produce $NH_3$ from natural gas as the $H_2$ source with a specific energy demand of around 28 GJ per ton of $NH_3$ and an overall process energy efficiency of up to 70% [28].

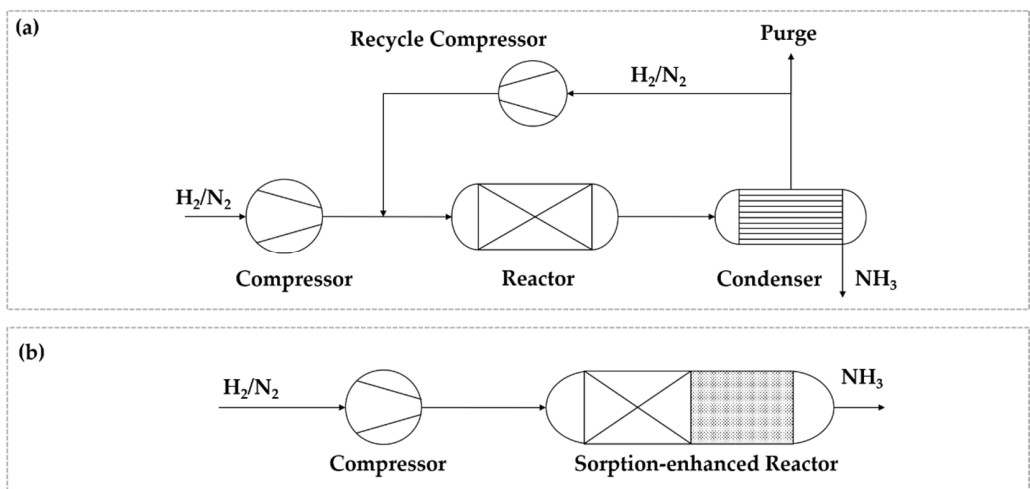

**Figure 1.** Simplified process diagram for the conventional Haber–Bosch process (**a**) and an integrated reactor concept with in situ removal of $NH_3$ in a sorptive process (**b**).

### 2.3. Power-to-Ammonia

The term PtA is used for $NH_3$ synthesis processes based on renewable energy, whereby the reactant $H_2$ is produced via water electrolysis, while $N_2$ is provided via air separation technologies. The production of $NH_3$ from renewable $H_2$ was successfully demonstrated on a pilot-scale [29]. An overview of existing and planned demonstration plants was given by Ayvali and co-workers [30]. The ongoing research into PtA technology can be categorized mainly into two pathways [31]. The first pathway describes the implementation of the conventional Haber–Bosch process loop into a process based on renewable $H_2$ feedstock. This concept is mainly discussed for the world-scale production of $NH_3$ and the reutilization of existing production plants with $H_2$ derived from renewable sources. The second pathway is research-driven and focuses on non-conventional and enhanced production technologies. The focus is on overcoming the described challenges of PtX processes to increase the process efficiency for decentralized $NH_3$ production units. In this context considered PI methods and their potential are described in the following sections.

### 2.4. Process Intensification Methods

The heterogeneously catalyzed reaction is the centerpiece of the synthesis process and defines the operation conditions for further unit operations. Therefore, the investigation towards highly active catalysts was part of the research and development in the course of the century of industrial $NH_3$ synthesis. This led to a high number of different active, promoter and support materials, which were investigated. Ruthenium-based catalysts were identified as a promising material for $NH_3$ synthesis under milder reaction conditions due to their high catalytic activity [32]. Recent studies showed that the synthesis of $NH_3$ at pressures below 10 bar and at temperatures between 300 and 400 °C was feasible [33,34], which provided a decrease in energy demand (i.e., for synthesis gas compression). Furthermore, a fast response and a high stability towards the thermal cycling was demonstrated [35]. This is necessary for dynamically operated $NH_3$ reactors for renewable feedstock conversion in remote locations. Accordingly, the usage of Ru-based catalysts and the associated shift towards mild reaction conditions is expected to be an important contribution for the aimed PI.

If $NH_3$ is synthesized at a lower pressure, the operating conditions for the separation of $NH_3$ changes. Since the partial pressure of $NH_3$ in the product gas decreases due to lower $NH_3$ equilibrium concentration and the lower total pressure, lower temperatures for condensation are necessary. Therefore, the energy demand for separation via condensation in low-pressure processes increases, while separation efficiency decreases [36]. Importantly, the complete separation of $NH_3$ is crucial for the process energy efficiency, as $NH_3$ remaining in the recycle stream leads to a decrease in the formation rate due to unfavorable equilibrium constraints [37]. To overcome this challenge, the replacement of condensation with adsorptive or absorptive $NH_3$ separation for low pressure processes was discussed as a promising option in various publications [38–47]. These techniques, which can be carried out at elevated temperatures, provide a selective separation of $NH_3$ and can reach a low residual $NH_3$ concentration in the recycle stream (<1%) [31]. Furthermore, the energy demand for the latter separation approach is rather low.

Various materials for the sorption of $NH_3$ are available, whereby the research focuses on zeolitic materials [40] and metal halide salts [48]. The main challenge arises from the fact that ad-/absorbed $NH_3$ needs to be released in an additional pressure or temperature swing process (PSA/TSA). The full release of the adsorbed $NH_3$ and the stability of the adsorbent under the periodic operation remain undergoing research topics for sorptive $NH_3$ separation [44]. A further promising PI method is the process integration of the reactor and a sorption-based separation process into a single multifunctional reactor, as described in Figure 1b. The in situ removal of the reaction product shifts the equilibrium towards the product site and mitigates the thermodynamic limitation of the synthesis even at a low pressure. Moreover, the sorptive in situ removal replaces the inefficient condensation of $NH_3$ at a low pressure. First approaches for the realization of a multifunctional reactor using a Ru-based catalyst for $NH_3$ synthesis and metal halide salts for separation were published recently [41].

The potential of the PI strategies discussed above are compared quantitatively with respect to the specific energy demand per ton of liquid $NH_3$ produced as published in the recent literature ([31,37,49]). Figure 2 shows the energy demands for synthesis and recycle gas compression, as well as for cooling in the condenser. A conventional Haber–Bosch process is compared with a low-pressure synthesis process, a process with sorptive separation and a process with in situ separation.

As discussed above, more active catalysts allow for lower operation pressure. The comparison therefore illustrates the effect of novel catalysts in conventional process concepts. For conventional processes operating at 300 bar, the specific energy demand is mainly caused by the compression of the synthesis gas and amounts to around 6 GJ/t [37]. The reduction of the pressure to 100 bar leads to a decrease in overall energy demand [50]. At mild pressures of 20 bar the energy demand surprisingly increases drastically, mainly driven by the energy intensive condensation of the produced $NH_3$ at lower temperatures

compared to conventional processes [49]. Replacing the condensation with an adsorptive separation downstream the $NH_3$ synthesis reactor allows for a significant decrease in energy demand at the low synthesis pressure of 20 bar [31,51]. For the integrated process with in situ $NH_3$ separation, a further reduction of energy demand down to around 3 GJ/t can be expected according to Smith et al. [49], including the energy demand for desorption via a temperature swing process and the liquefaction of desorbed ammonia. This integrated process clearly outperforms the conventional one. The main reason is the shift in equilibrium constraints provided by the high catalyst activity at low temperatures combined with the in situ removal of the reaction product $NH_3$. Thereby, a higher conversion at mild synthesis conditions is achieved, which reduces or even avoids the recycle of unreacted $N_2$ and $H_2$ and consequently the energy demand for compression and separation. Additionally, the plant size and number of apparatuses decreases, which paves the way for less complex production plants. Importantly, the loss of reactant gases via purging is avoided, if the recycle of unreacted compounds is obsolete.

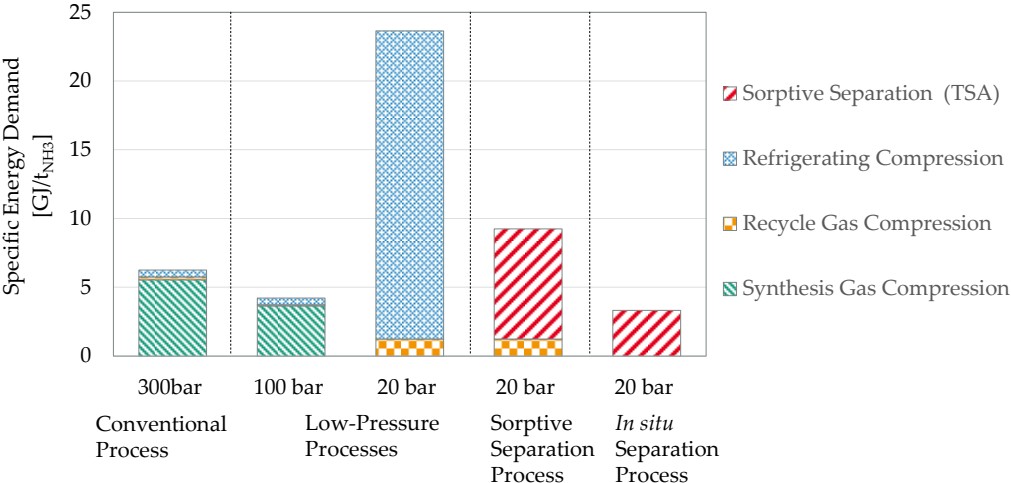

**Figure 2.** Comparison of specific energy demand per ton liquid $NH_3$ of the conventional synthesis process with novel low-pressure concepts using different separation strategies.

It must be pointed out that the concept of in situ removal requires low operation temperatures in order to facilitate the $NH_3$ ad-/absorption and hence, highly active catalysts are a prerequisite for this strategy. Further research, however, is required prior to the implementation of the intensified process concept. Besides the selection of sufficiently stable and selective sorbent materials for $NH_3$ removal, the reactor and process concept need to be developed considering the reaction and the subsequent adsorption and desorption steps. Regarding the reactor design, the spatial arrangement of catalyst and sorbent needs to be explored. With respect to the process concept the operation mode needs to be developed, which enables a stable reactor operation and product composition considering the inherently dynamic ad- and desorption steps.

Interestingly, the described PI method of in situ removal combines three strategies. The utilization of highly active catalysts, the replacement of the separation technology, and the integration of two unit operations into one single apparatus, in order to fully exploit the potential of PI. Considering the stated challenges of PtX and the potential of PI, several benefits of the described process approach are identified: The quantitative comparison of the specific energy demand exhibits an increase in energy efficiency, which represents one goal of the PI measures [11]. Regarding the production scale, the presented PI methods offer a promising perspective. As the presented energy consumption for the conventional process was evaluated for large scale production of ca. 330 t/d and reported to decrease at smaller scales [31], the in situ PI approach is yet considered for small production capacities. The desired reduction of plant size and increased flexibility of the resulting process concept can only be discussed qualitatively in the current analysis. Yet, the discussed reduction

of reactor pressure and temperature contribute positively towards the realization of the desired dynamic reactor operation. Together with employing more active catalysts, these synthesis conditions and simplification measures remain important PI outcomes that can yield a feasible process under dynamic conditions.

### 3. Process Intensification for Power-to-DME Processes

*3.1. Background*

To date DME is produced at large scale with a total world production capacity of approximately 5 Mt/a. The major application is the replacement and blending of Liquefied Petroleum Gas (LPG), mainly in China [52]. Furthermore, DME is used as a propellant, solvent and intermediate for subsequent syntheses of important end products [53]. Moreover, DME is under investigation as a diesel substitute [53]. The conventional DME production route refers to the so-called indirect two-step route presented in Figure 3a. In the first step, MeOH is produced from synthesis gas—a mixture of $H_2$, CO and $CO_2$—which can be produced based on fossil or renewable feedstock. The crude MeOH—a MeOH/water mixture—is subsequently separated from unreacted syngas by flash separation. The unreacted syngas is recycled, while the crude MeOH is purified by means of distillation, to remove water and low-boiling components formed in the MeOH synthesis [54]. For the following DME synthesis step, the purified MeOH is evaporated, pre-heated and fed into a fixed-bed reactor equipped with a solid acid catalyst. DME is formed by dehydration of MeOH in a heterogeneously catalyzed gas-phase reaction at temperatures between 220–360 °C and pressures up to 20 bar according to the stoichiometric Equation (2) [53,55].

$$2\,CH_3OH \rightleftharpoons CH_3OCH_3 + H_2O$$
$$\Delta H^o_{298\ K} = -23.5\ kJ\ mol^{-1} \tag{2}$$

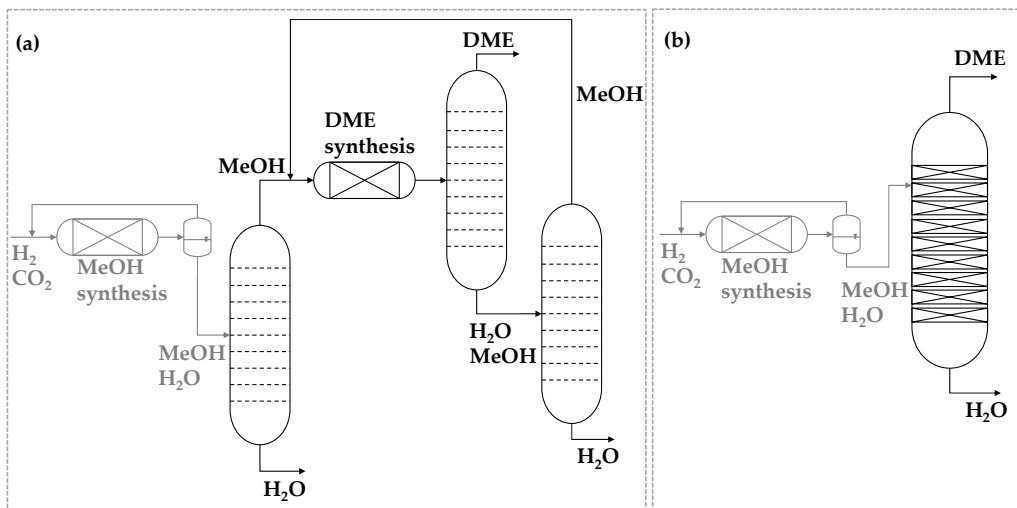

**Figure 3.** Simplified process flowsheet for the conventional DME production process (**a**) and the PI approach by reactive distillation (**b**).

The most widely used catalyst for MeOH dehydration is $\gamma$-$Al_2O_3$ due to its low cost, high selectivity, high specific surface area and good mechanical and thermal stability [56]. The conversion per pass is typically around 70–85%, which is already in the vicinity of chemical equilibrium. As the activity of alumina is strongly affected by water, purified MeOH needs to be used as feedstock with low amounts of residual water. The reaction product consists mainly of unreacted MeOH, DME and water, which are separated conventionally by two-step distillation. Water is removed in a first distillation column, while the remaining MeOH-DME mixture is fed to a second distillation column, where pure DME is obtained as distillate and MeOH as bottom product. The unreacted MeOH is recycled to the DME synthesis reactor [53].

### 3.2. Process Intensification Methods

One major drawback of the conventional DME production process is an incomplete conversion, resulting in the necessity of an energy-intensive purification process and a recycle stream. Consequently, many PI methods focus on strategies to shift the thermodynamic equilibrium in the reaction through removal of the byproduct water. Whereby it can be distinguished between sorption-based, membrane-based and distillation-based (reactive distillation) in situ water-removal.

In the sorption-based PI method, water is removed in situ by selective adsorption on adsorbent particles (i.e., zeolites) mixed with the catalyst particles [57]. While this concept allows a simple reactor design, the inherent transient adsorption and desorption processes require cyclic operation of multiple reactors in parallel to achieve a continuous synthesis [58].

The membrane-based PI method in contrast leads to a compact integrated reaction and separation unit while allowing a continuous operation by a permanent water-flux through the membrane [59]. In practice, however, the multi-objective demands for the membrane remain hard to fulfill. On the one hand, the membrane must be highly selective towards the permeation of water but shall not be permeable for MeOH. On the other hand, the membrane must be resistant to high pressure differences. For that purpose, hydrophilic zeolite membranes represent promising materials, but the fabrication process, the scalability and stability are limiting factors for commercialization [60].

The reactive distillation (RD) approach is a PI method based on the combination of reaction and distillation within a single apparatus, with the aim of simplification for the continuous DME production process without the need for expensive or sensitive materials and components. It is the PI method of focus in this work and discussed in further detail in the following section.

### 3.3. DME Synthesis by Reactive Distillation

The MeOH dehydration to DME presents a suitable reaction for RD due to three main reasons:

(a)    the reaction is limited by chemical equilibrium,
(b)    the reaction is exothermic, which allows the utilization of the reaction enthalpy to reduce the reboiler heat demand,
(c)    and the components MeOH, DME and water exhibit a high relative volatility, thus allowing a good thermal separation capability.

In the RD approach illustrated in Figure 3b, the crude MeOH in liquid state is fed to the RD column at the top of the reactive section and flows downwards. Contrary to the conventional DME synthesis, MeOH is dehydrated in the liquid phase, catalyzed by a solid acidic catalyst, which is fixed in a structured catalytic packing inside the reaction section of the column. The formed DME exhibits a higher vapor pressure than MeOH and rises upwards as vapor. Since water is less volatile than MeOH, it consequently concentrates in the bottom section of the column. Thereby, both products are separated from each other, which favors product formation from a thermodynamic perspective. By adjusting the design parameters of the RD unit, full conversion of MeOH can be achieved and hence the only product streams are a pure DME distillate at the top and pure water at the bottom. This RD process can significantly simplify the conventional reaction-separation-recycle sequence as depicted in Figure 3.

The simplified process design can be accompanied by an increased complexity regarding the control and instrumentation of the column. However, the main challenge for DME synthesis via RD is that a countercurrent flow of a liquid and a gas phase is inherent, which renders the reaction conditions as contrary to the conventional gas phase synthesis. For this reason, the reaction temperature is limited by the evaporation temperature at the desired column pressure and therefore, considerably lower than the conventional synthesis reaction temperature in fixed bed reactor. Thus, the rate of the liquid-phase reaction is relatively low, which leads to larger amounts or significantly more active catalysts required

for the desired productivity. Furthermore, full conversion of the MeOH feed is necessary to fully exploit the RD potential. Consequently, the catalyst performance represents the key parameter for the required RD column size. Since conventional catalysts, such as $\gamma$-$Al_2O_3$ and zeolites, exhibit a very low activity under the RD operational conditions, suitable catalysts are required. Acidic ion exchange resins (IER) are among the most promising candidates. Current literature investigations are limited to the application of Amberlyst 35, which dictates a temperature limit of 150 °C, corresponding to a column pressure of roughly 11 bar [61–63]. Catalysts providing a higher thermal stability present a major opportunity for a significantly higher reaction rate according to the Arrhenius law. However, higher reaction temperatures require higher column pressures, resulting in higher investment costs, as well. Additionally, the reflux ratio of the column bears a strong influence on the required amount of catalyst This interplay between catalyst selection and process design holds large potential for process optimization of the RD process and is the key for tuning this PI approach towards two possible PI targets: On the one hand, the process can be optimized towards a minimal plant size, which in return means sacrificing some energy efficiency potential. On the other hand, the process can be optimized towards maximum energy efficiency -an important prerequisite in PtX process context-, resulting in a non-optimized plant footprint.

Figure 4 compares the specific energy demand per ton DME of the process intensification approach based on RD with the conventional process. In both process routes, the residual heat from the exothermic MeOH synthesis is accounted for as a negative energy demand. The diagram illustrates that the thermal energy demand for MeOH distillation and evaporation as well as both distillation columns amount to more than 4.7 GJ/t for the conventional DME process. Considering integration of the reaction enthalpy of the MeOH synthesis 2.12 GJ/t is still required to be supplied from external sources. The RD approach, in contrast, allows omission of most of the required energy demands since the respective unit operations are not required for the process. Hence, the RD reboiler remains the only major heat consuming equipment. As mentioned above, the absolute energy demand depends on the complex interplay between catalyst selection and sizing of the column and is adjustable over a wide range. Overall, even a net-zero energy demand process could be realized based on the RD approach.

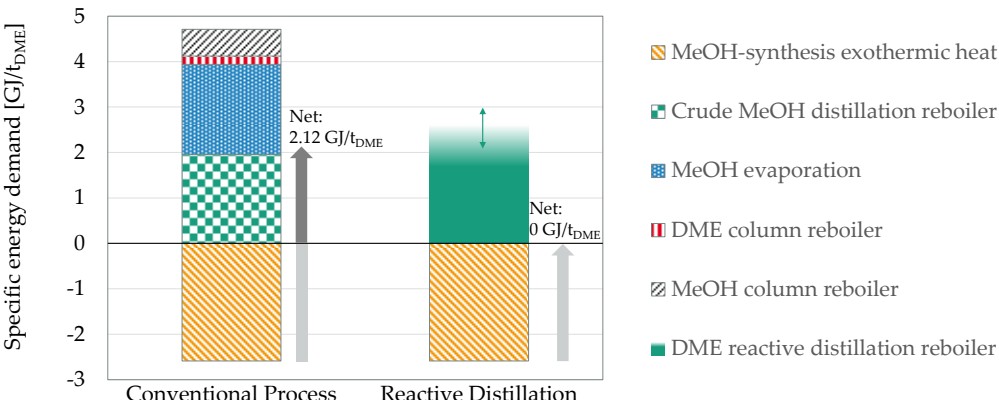

**Figure 4.** Comparison of the specific energy demand per ton DME between the conventional DME process and the RD approach; own calculations based on [57,60].

This comparison illustrates how the presented PI approach comprises multiple intensification techniques, all of which fall within the category of "process-intensifying methods". Firstly, the shift from gas- to liquid-phase reaction eliminates the necessity of the energy-intensive evaporation of the feed, resulting in energy savings and a process simplification. Besides, the implementation of the RD concept represents a process integration of three unit operations into one multifunctional reactor:

(a)    the feedstock purification (crude MeOH distillation),

(b)    the DME synthesis reactor and

(c)    the product separation.

This exceptionally high degree of intensification is possible since water is the by-product of both the DME synthesis and the methanol synthesis. Hence, both reaction steps benefit from the in situ water removal in the RD approach at the same time. In practice, the integration of the feedstock purification (crude MeOH distillation) represents a major advantage, as this process-step remains the main energy consumer in the conventional process consuming a share of more than 40% of the overall energy demand of DME production starting from crude MeOH. Furthermore, the in situ removal of the product allows the complete conversion of the feedstock and consequently avoids recycle streams including corresponding equipment, which remains common in multifunctional reactors. This achievement is of great importance regarding a self-sufficient operation in remote areas with high PtX potential as explained in the introduction.

Overall, the intensified process shows a lower utilities demand and a significantly simplified process layout compared to the conventional process, and thus illustrates how multiple process-intensifying methods can be combined to yield a better process.

## 4. Process Intensification for Power-to-OME Processes

### 4.1. Background

In comparison to $NH_3$ and DME, the current global production capacity of OME is relatively small and is mainly in China. Due to the chemical and physical properties coupled with the non-toxic, environmentally benign, and favorable combustible behavior, OME can be used in a wide range of applications. It was investigated as a selective polymer solvent [64], for $CO_2$ absorption [65], and as a fuel in fuel cells [66–70]. $OME_n$ (chemical formula $CH_3O(CH_2O)_nCH_3$) with the chain length $3 \leq n \leq 5$ ($OME_{3–5}$) particularly is intensively investigated for the combustion application as a diesel blend or substitute. This is due to significantly low soot formation upon combustion, while allowing significant reduction of $NO_x$ emissions [64–75].

$OME_{\geq 2}$ ($n \geq 2$) is synthesized in the liquid phase at 50–100 °C in presence of a solid acid catalyst (e.g., IER). Applying MeOH as the methyl capping group supplier and a formaldehyde (FA, chemical formula $CH_2O$) source, $OME_{\geq 1}$ ($n \geq 1$) and $H_2O$ are formed, following Equation (3). Using methylal ($OME_1$, $n = 1$) or DME instead of MeOH, no $H_2O$ is formed as by-product, as shown in Equation (4). Formalin and para-FA (pFA, chemical formula $HO(CH_2O)_nH$ with $8 \leq n \leq 100$) can be used as the FA source, but already contains $H_2O$ whose presence leads to the formation of several side-products. As an alternative FA source, trioxane (TRI, chemical formula $(CH_2O)_3$) or anhydrous FA ($FA_{an}$) can be applied as water-free FA sources. Thereby, TRI is converted to FA in presence of an acid catalyst following Equation (5) [76,77].

$$2\,CH_3OH + n\,CH_2O \rightleftharpoons CH_3O(CH_2O)_nCH_3 + H_2O \qquad (3)$$

$$CH_3O(CH_2O)_{n-1}CH_3 + CH_2O \rightleftharpoons CH_3O(CH_2O)_nCH_3 \qquad (4)$$

$$\Delta H^o_{298\,K} = -25.2\ kJ\,mol^{-1}$$

$$(CH_2O)_3 \rightleftharpoons 3\,CH_2O \qquad (5)$$

$OME_n$ can be produced following an aqueous route under presence of $H_2O$ or an anhydrous route, depending on the choice of feedstock [77]. Typical for the anhydrous route is the synthesis from $OME_1$ and TRI, which exhibits the benefit of a simple product purification due to the absence of $H_2O$ within the synthesis product, but requires expensive feedstock [77]. An alternative concept is the aqueous route, which is based on MeOH and concentrated formalin, which is a significantly less expensive feedstock, but requires a complex product purification due to the presence of $H_2O$ within the synthesis mixture. The purification of the highly non-ideal and reactive product mixture is cumbersome, due to several azeotropes, complex vapor-liquid-liquid equilibria (VLLE), challenges regarding

FA solidification and the separation of $H_2O$ from the process. $H_2O$ management remains a key hurdle for the realization of a scalable $OME_{3-5}$ production. Research and development focused in the last decades on various $H_2O$ separation strategies, such as extraction [78–85], adsorption [76,86], or membranes [87]. The long-term stability of the materials applied for the previous concepts, as well as the efficiency and the scalability of these approaches are crucial for their application and remain under investigation.

Besides the $H_2O$ separation, the synthesis towards OME is less selective for the aqueous route, due to the formation of many side-products which reflects upon the process energy efficiency [88]. $H_2O$ is not only formed during the synthesis but enters the process with the FA source formalin. This is a result of the state-of-the-art aqueous FA ($FA_{aq}$) production processes, which are based on the partial oxidation of MeOH with air (Equation (6)) and dehydrogenation of MeOH (Equation (7)). Silver is used as a catalyst in the BASF process and iron-molybdenum based catalysts are used in the Formox process [89].

$$CH_3OH + \tfrac{1}{2}O_2 \rightarrow CH_2O + H_2O$$
$$\Delta H^o_{298\text{ K}} = -159 \text{ kJ mol}^{-1} \tag{6}$$

$$CH_3OH \rightleftharpoons CH_2O + H_2$$
$$\Delta H^o_{298\text{ K}} = 84 \text{ kJ mol}^{-1} \tag{7}$$

$$H_2 + \tfrac{1}{2}O_2 \rightarrow H_2O$$
$$\Delta H^o_{298\text{ K}} = -243 \text{ kJ mol}^{-1} \tag{8}$$

*4.2. Power-to-OME*

OME production is based on MeOH, which can be produced from renewable feedstock such as green $H_2$ and $CO_2$ [54,90]. While a large-scale production of MeOH based on renewable feedstock has already been implemented [91], only a paucity of information remains available regarding OME production plants. In China, OME plants are reported to be in operation or under construction with production capacities of 10–400 kt/a but are mostly based on fossil feedstock. However, no information is available regarding the product quality, reproducibility and long-term production capacities [92,93]. The bottleneck for the technology deployment for OME production remains the implementation and demonstration of a simple, efficient and scalable product purification process.

*4.3. Process Intensification Methods*

To circumvent the separation of $H_2O$ from the process, different process modifications were proposed adopting methyl capping group suppliers (e.g., DME or $OME_1$) and using TRI or pFA as FA sources [92,94,95]. This leads to two main Power-to-OME process concepts based on green $H_2$ and captured $CO_2$ which are depicted in Figure 5. While Figure 5a is based on formalin as FA source, Figure 5b represents the concept using anhydrous FA ($FA_{an}$) instead. From methanol and $FA_{an}$ $OME_1$ is produced to be reacted further with $FA_{an}$ for higher OME synthesis. Therefore, no $H_2O$ is formed within the OME synthesis in this route [90].

While the solid pFA presents a cheaper feedstock than TRI, it still contains about 1–10 wt.% $H_2O$ [89], which requires removal from the process. The alternative $FA_{an}$ feedstock can be synthesized via endothermic dehydrogenation of MeOH at temperatures > 650 °C following Equation (7) and avoids $H_2O$ formation, while $H_2$ is formed as valuable side product [96–98]. The implementation of the $FA_{an}$ synthesis into the OME process chain starting from the production of MeOH (Figure 5b) contains two main benefits. Firstly, the provision of FA without any additional $H_2O$ and without the energy intensive production of TRI, as discussed above. The second benefit addresses the potential for reutilizing the valuable side product $H_2$. Downstream to the separation of $FA_{an}$ from the gaseous product stream, $H_2$ can be separated and used for the production of MeOH, as seen in (Figure 5b). Therefore, instead of oxidizing this valuable $H_2$ to $H_2O$ following the state-of-the-art $FA_{aq}$ synthesis (Equation (6)), the reutilization of $H_2$ following the $FA_{an}$ synthesis (Equation (7))

lowers the amount of $H_2$ required to produce the target end product $OME_{3-5}$. This reflects significantly on the operational costs, especially considering the context of PtX processes using green $H_2$ [90].

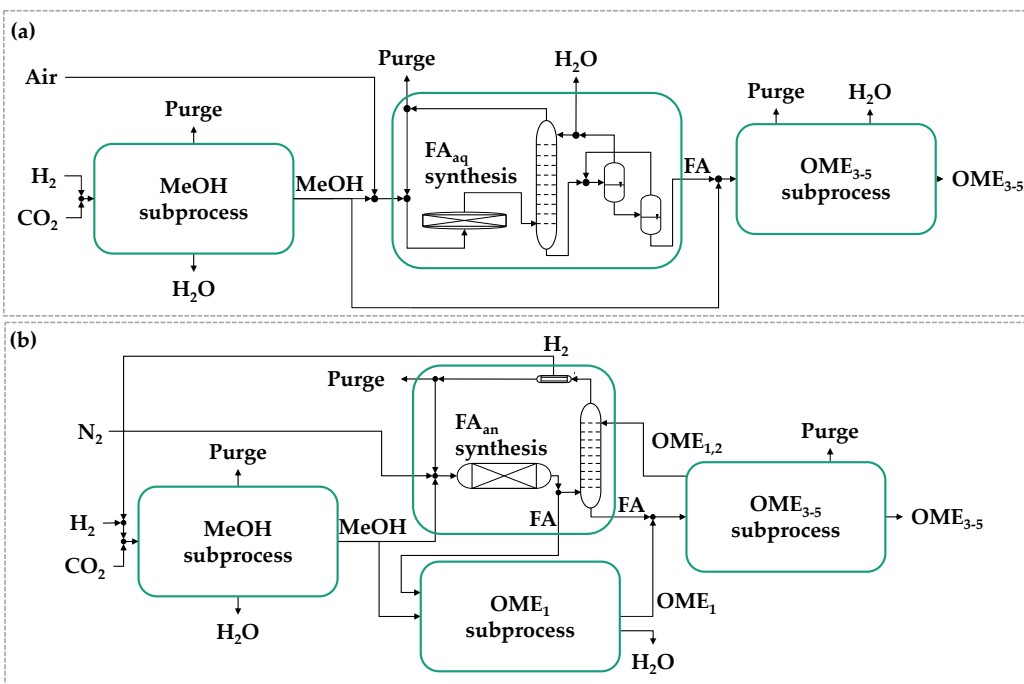

**Figure 5.** Process concepts for $OME_{3-5}$ production from MeOH and concentrated formalin (**a**) and from MeOH and anhydrous FA (**b**).

The $FA_{an}$ synthesis was investigated and experimentally tested over a broad range of catalyst materials by several research groups, but was not yet demonstrated in long-term experiments or in an industrially relevant environment [96–98]. Besides the catalyst stability as a key challenge to realize the $FA_{an}$ synthesis, a suitable reactor design considering the endothermic reaction at a high temperature and the strongly reducing $H_2$ environment remains challenging.

Moreover, a complete conversion of MeOH is required and a high selectivity to FA is desired to circumvent the $H_2O$ separation management for the OME product purification. Sauer et al., [97] achieved a complete conversion of MeOH with a selectivity to FA of 70% and the formation of CO as a side product using an electrically heated tube wall reactor with a catalyst coated inner wall. High MeOH conversion and FA selectivity require challenging reaction conditions: For high selectivity, in particular, very short residence times in the heated reaction zone and on the catalyst surface, as well as rapid quenching of the product stream to 100–150 °C is required to avoid dissociation reactions towards CO. The temperature should not fall below 100 °C to avoid potential solidification of gaseous monomeric FA [97]. Ouda et al., [98] developed an electrically heated annular counter current reactor (ACCR) for the $FA_{an}$ synthesis achieving a rather high FA selectivity of 90% at low MeOH conversions of 40%.

The potential of the choice of the FA source - being either concentrated formalin or $FA_{an}$ - is compared with respect to the specific energy demand, the carbon footprint and the specific wastewater formation based on the results of Mantei et al. [90]. Figure 6 compares the specific energy demand per ton $OME_{3-5}$ produced depending on the FA source, achieved after heat integration between all subprocesses. The overall energy demand is distinguished in electricity, steam and heat (above 250 °C), while the $H_2$ required in the processes is also accounted for as an energy demand based on the lower heating value. The process based on concentrated formalin requires less electricity, steam and heat than its counterpart based on $FA_{an}$. The $FA_{an}$ based process, in contrast, requires high amounts of

heat, due to the endothermic reaction at >650 °C. However, $FA_{an}$ exhibits a slightly lower overall energy demand, due to 20% less $H_2$ required as a feedstock. Hence, both processes are rather comparable exhibiting energetic efficiencies in the order of 50–54%.

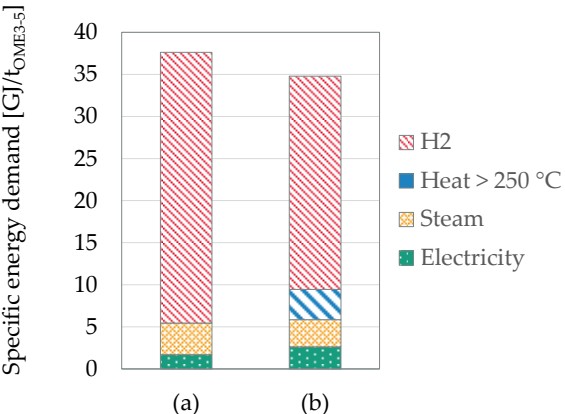

**Figure 6.** Comparison of the specific energy demand per ton $OME_{3-5}$ between the process based on concentrated formalin (**a**) and on $FA_{an}$ (**b**).

Besides the energy demand, the reduction of greenhouse gas emissions represents an important motivation for implementing PI approaches [13]. In terms of carbon footprint the $OME_{3-5}$ production using $FA_{an}$ clearly outperforms the formalin-based process, since the heat required for the endothermic reaction is provided directly via renewable electricity [90]. Comparing the specific amount of wastewater produced for the production of $OME_{3-5}$ the $FA_{an}$ based process (1 kg/kg) outperforms its counterpart using formalin (1.3 kg/kg), as well. Furthermore, the wastewater produced in the formalin route still contains about 10–15 wt.% of FA, which complicates its treatment. Therefore, the $OME_{3-5}$ production via $FA_{an}$ combines a slightly higher energetic efficiency with a more sustainable production of $OME_{3-5}$, which remains within the confines of the main goals of PI methods in the PtX context.

With innovation and developments towards electrically driven selective $FA_{an}$ synthesis reactor, the intensified process shows a significant potential towards simpler, compacter, more energy efficient, and importantly a low carbon footprint overall OME value chain.

## 5. Summary and Conclusions

In this work, the potential of PI methods for the development of efficient, competitive, compact and low-maintenance thermochemical processes was identified and discussed supported by examples for the three PtX-products namely: $NH_3$, DME and OME. A brief description of the-state-of-the-art processes and PI approaches was provided. In the case of $NH_3$, advanced ruthenium catalysts allowed lower operating temperatures, thus paving the way for adsorption as a new separation technology, which can be integrated into a multifunctional reactor with in situ removal of $NH_3$. Consequently, the specific energy demand for the integrated reactor presented as per ton of $NH_3$ could be almost halved in comparison to the conventional Haber–Bosch process. In the case of DME synthesis, ion exchange resin catalysts allow a reduction of the reaction temperature below the boiling temperature of methanol, thus allowing the shift from gas- to liquid-phase synthesis, which in turn enables the implementation of an RD process. A quasi-net-zero energy demand process could be achieved in a single unit operation replacing three unit operations when the integration between MeOH synthesis and DME–RD is properly realized. For PI of the OME synthesis process, the $FA_{an}$ synthesis based on methanol dehydrogenation in electrically heated reactors was implemented instead of the state-of-the-art $FA_{aq}$ synthesis. Furthermore, $OME_1$ was used as a methyl capping group supplier in place of MeOH which together circumvents the challenge of the cumbersome $H_2O$ separation within the conventional OME production process. Another benefit observed was the separation and

recycling of the valuable by-product $H_2$ of the $FA_{an}$ synthesis. This approach led to 20% less $H_2$ feedstock required per ton of $OME_{3-5}$ in comparison to the conventional aqueous process. Additionally, approximately 23% less wastewater was produced based on the $FA_{an}$ synthesis.

Considering the described challenges of PtX processes, the PI measures presented offer several promising solutions. Primarily, process integration measures allow for significantly simplified processes, consequently leading to reduced component numbers and reduced maintenance efforts. Moreover, the potential elimination of recycle loops due to equilibrium shift of the reaction towards products reduces the complex interactions in a dynamic operation. Besides, simpler and smaller recycle loops can reduce maintenance efforts of the circulating equipment i.e., compressors. Additionally, the PI methods offer the potential to reduce the energy demand of the previously discussed processes against the conventional ones. All the PI approaches extended and discussed are research endeavors which remain under development in our work group illustrating the relevance of PI approaches in PtX processes.

**Author Contributions:** Introduction, Summary and Power-to-Ammonia, T.C.; Power-to-DME, M.S.; Power-to-OME, F.M.; writing—review and editing, O.S. and R.G.; scientific supervision, O.S.; project administration, O.S. All authors have read and agreed to the published version of the manuscript.

**Funding:** This research received no external funding.

**Institutional Review Board Statement:** Not applicable.

**Informed Consent Statement:** Not applicable.

**Data Availability Statement:** Analyzed data available in the original sources. See citations.

**Acknowledgments:** Deutsche Bundesstiftung Umwelt (DBU) is gratefully acknowledged for funding of the work of Thomas Cholewa (20020/671), Franz Mantei (20018/541) and Malte Semmel (20020/662).

**Conflicts of Interest:** The authors declare no conflict of interest.

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
