# Peer review of "Process Intensification Strategies for Power-to-X Technologies"

_2305-7084, doi:10.3390/chemengineering6010013_

Round 1

Reviewer 1 Report

General Comments

- Academic interest of the topic: Very Good

- Industry interest of the topic: Very Good

- Originality of the topic: Good

- Quality of the abstract: Good

- List of keywords: Good

- Presentation: - Originality: Very Good

                        - Clarity: Very Good

                        - Concision: Good

- Figures: - Number (6) Very Good

                 - Clarity: Very Good                               

- Quality of the discussions: Very Good

- Quality of the conclusions: Very Good (but the length of the text should be reduced)

- List of References: Very Good

- Adaptability of the article for a “Chemengineering” issue: Very good

The field of the investigation concerns a very important academic and industry topic, i.e. Process Intensification (PI) involved in the production of green Hydrogen using renewable energy and the subsequent synthesis of chemicals in Power-to-X (PtX) processes. The potential of PI methods combining heterogeneously catalyzed reactions and the subsequent separation processes aiming to shift the thermodynamic equilibrium towards the desired product is identified and discussed for three important products, namely Ammonia, Dimethyl Ethers (DME) and Oxymethylene Dimethylethers (OME). For these PtX-products, Process Intensification is focused on process-intensifying methods rather than process-intensifying equipment or technologies.

This investigation enters in the general context of recognizing Hydrogen and PtX products as major levers to permit the ramping up of renewable energy capacities and the consequent use of green electrons in different industry sectors. Thus Process Intensification approaches may offer the maximum utilization of the valuable renewable feedstock and the simpler production processes by integrating unit operations to overcome thermodynamic limitations and also in some cases to eliminate the recycle loops. Different PI strategies approaches have been clearly developed by the authors for the three important PtX produced previously mentioned.

This is a very interesting paper which certainly deserves to be published in “Chemengineering”. The work background is scientifically sound and the investigation is well presented. However,  the length of the manuscript is large (the reader is very happy when at last he reaches the end of the manuscript!), and it deserves to be reduced in taking into account the few following detailed comments.  

Detailed Comments

 - The PtX products: Ammonia, Dimethyl Ethers (DME) and Oxymethylene Dimethylethers (OME) should be mentioned within the text of the Abstract,

- “Reactive Distillation”, “Adsorptive or absorptive Separation” should be added in the list of Keywords,

-  In the text, “Error! Reference source not found” is mentioned at least 10 times. The authors should precise or fulfill these missing references,

- The lengths of the chapters “introduction” and “summary and conclusions” should be reduced by a factor 2.

Author Response

Fraunhofer ISE  |  Heidenhofstrasse 2  |  79110 Freiburg

Manuscript ID: ChemEngineering-1525293

Response to Reviewer Comments on submitted Paper “Process Intensification strategies for Power-to-X Technologies”.

Dear ChemEngineering Editorial Team and Reviewers,

On behalf of my co-authors and I, please find via the Manuscript central submission portal, the resubmission of our article entitled “Process Intensification strategies for Power-to-X Technologies. We are happy to receive the positive comments and suggestions of the reviewers, who have assisted in improving the quality of this paper.

====================================================================

Comments to the Author: This is a very interesting paper which certainly deserves to be published in “Chemengineering”. The work background is scientifically sound and the investigation is well presented. However, the length of the manuscript is large (the reader is very happy when at last he reaches the end of the manuscript!), and it deserves to be reduced in taking into account the few following detailed comments. 

1) The reviewer suggests that the PtX products: Ammonia, Dimethyl Ethers (DME) and Oxymethylene Dimethylethers (OME) should be mentioned within the text of the Abstract

Author’s Response: The PtX Products are mentioned within the abstract in the submitted and the revised versions

2) The reviewer suggests “Reactive Distillation”, “Adsorptive or absorptive Separation” should be added in the list of Keywords,

Author’s Response: Theses Keywords are added to the list of keywords

3) In the text, “Error! Reference source not found” is mentioned at least 10 times. The authors should precise or fulfill these missing references,

Author’s Response: References have been reworked carefully and missing references have been added

4) The lengths of the chapters “introduction” and “summary and conclusions” should be reduced by a factor 2.

Author’s Response: The aim of this work is to highlight the arising challenges concerning PtX technologies and give impression about the potential of some PI methods to tackle these challenges and rather improve these processes. The respected suggestion was considered and The referred chapters have been reviewed carefully and shortened whenever possible. From authors perspective, the mentioned chapters are serving their objectives giving a satisfactory overview and understanding of both topics is necessary.

====================================================================

We hope the editorial team and reviewers find our response to their satisfaction, and we would like to thank the reviewers again for their time and indeed their suggestions regarding the improvement of our submitted manuscript, and accordingly we look forward to your feedback.

With my best regards,

Dr.-Ing. Ouda Salem

On behalf of my co-authors Thomas Cholewa, Malte Semmel, Franz Mantei and Robert Güttel

Reviewer 2 Report

please review the sugestions and comments into the file (22 items)

Author Response

Dear ChemEngineering Editorial Team and Reviewers,

On behalf of my co-authors and I, please find via the Manuscript central submission portal, the resubmission of our article entitled “Process Intensification strategies for Power-to-X Technologies. We are happy to receive the positive comments and suggestions of the reviewers, who have assisted in improving the quality of this paper.

In the attached file is a point by point respinse to the valuable comments. 

With my best regards,

Dr.-Ing. Ouda Salem

On behalf of my co-authors Thomas Cholewa, Malte Semmel, Franz Mantei and Robert Güttel

Reviewer 3 Report

My comments have been addressed.

Author Response

Dear ChemEngineering Editorial Team and Reviewers,

On behalf of my co-authors and I, please find via the Manuscript central submission portal, the resubmission of our article entitled “Process Intensification strategies for Power-to-X Technologies. We are happy to receive the positive comments and suggestions of the reviewers, who have assisted in improving the quality of this paper.

A detailed point by point response to the valuable review comments is attached. 

With my best regards,

Dr.-Ing. Ouda Salem

On behalf of my co-authors Thomas Cholewa, Malte Semmel, Franz Mantei and Robert Güttel

Reviewer 4 Report

In this manuscript, the authors briefly reviewed the challenges of the PtX processes using different PI strategies, with their potential to overcome those challenges in selected PtX products. The study is timely to the field and would attract wide readership globally if published. However, a few things need to be addressed to put this article in perfect shape for publication.

  1. The abstract contains too many background and motivational stories. Please revise adequately to include the main areas this review covers, novelty, and how the findings from this review will be beneficial to the scientific community.
  2. Describe or clearly explain "PtX processes" in the introduction
  3. Sections 1.3 and 1.2 provide background and objective of the subject matter, therefore they should not be separated from 1. Introduction. They should form part of section 1 in that order.
  4. section 2.2, fix the "error reference not found"
  5. Include recent literatures on process intensification like https://doi.org/10.1016/j.coche.2018.12.006, https://doi.org/10.1016/j.rser.2021.111241 and https://doi.org/10.1016/j.rser.2012.05.014 to section 4.3 to get this study updated with recent scholarship.
  6. Too many errors with in-text referencing which could be from the reference manager used. Please correct this throughout the manuscript. 
  7. Section 5 is too long. Revise to capture major conclusions from this study, include some future prospects.

Author Response

(The authors gave the same response as above.)

Round 2

Reviewer 2 Report

the answers to suggestions and comments are adequate. The paper could be published.